# Harnessing the Potential of Biosurfactants for Biomedical and Pharmaceutical Applications

**DOI:** 10.3390/pharmaceutics15082156

**Published:** 2023-08-18

**Authors:** Chiara Ceresa, Letizia Fracchia, Andrea Chiara Sansotera, Mayri Alejandra Díaz De Rienzo, Ibrahim M. Banat

**Affiliations:** 1Department of Pharmaceutical Sciences, Università del Piemonte Orientale “A. Avogadro”, 28100 Novara, Italy; chiara.ceresa@uniupo.it (C.C.); letizia.fracchia@uniupo.it (L.F.); andrea.sansotera@uniupo.it (A.C.S.); 2School of Science, Engineering and Environment, University of Salford, Manchester M5 4WT, UK; m.a.diazderienzo@salford.ac.uk; 3Pharmaceutical Science Research Group, Biomedical Science Research Institute, Ulster University, Coleraine BT52 1SA, UK

**Keywords:** biosurfactants, antibacterial/antifungal/antiviral activity, antibiofilm agents, antiadhesive-coatings, anticancer agents, immunomodulatory activity, wound-healing promoters, drug delivery systems, marine producers, LAB

## Abstract

Biosurfactants (BSs) are microbial compounds that have emerged as potential alternatives to chemical surfactants due to their multifunctional properties, sustainability and biodegradability. Owing to their amphipathic nature and distinctive structural arrangement, biosurfactants exhibit a range of physicochemical properties, including excellent surface activity, efficient critical micelle concentration, humectant properties, foaming and cleaning abilities and the capacity to form microemulsions. Furthermore, numerous biosurfactants display additional biological characteristics, such as antibacterial, antifungal and antiviral effects, and antioxidant, anticancer and immunomodulatory activities. Over the past two decades, numerous studies have explored their potential applications, including pharmaceuticals, cosmetics, antimicrobial and antibiofilm agents, wound healing, anticancer treatments, immune system modulators and drug/gene carriers. These applications are particularly important in addressing challenges such as antimicrobial resistance and biofilm formations in clinical, hygiene and therapeutic settings. They can also serve as coating agents for surfaces, enabling antiadhesive, suppression, or eradication strategies. Not least importantly, biosurfactants have shown compatibility with various drug formulations, including nanoparticles, liposomes, micro- and nanoemulsions and hydrogels, improving drug solubility, stability and bioavailability, and enabling a targeted and controlled drug release. These qualities make biosurfactants promising candidates for the development of next-generation antimicrobial, antibiofilm, anticancer, wound-healing, immunomodulating, drug or gene delivery agents, as well as adjuvants to other antibiotics. Analysing the most recent literature, this review aims to update the present understanding, highlight emerging trends, and identify promising directions and advancements in the utilization of biosurfactants within the pharmaceutical and biomedical fields.

## 1. Introduction

The discovery of new bioactive molecules holds significant importance for combating numerous diseases and infections. Among them, in recent years, research has been focused particularly on the identification of molecules of natural origin [1,2,3,4]. Biosurfactants (BSs), or tensio-active biomolecules, are the product of the biosynthesis of a great number of different microorganisms, from bacteria (*Acinetobacter*, *Bacillus* and *Pseudomonas*) to filamentous fungi (*Aspergillus*, *Fusarium*, *Penicillium*, *Trichoderma*, *Ustilago*) and yeast (*Kluyveromyces*, *Pseudozyma*, *Rhodotorula*, *Torulopsis*, *Saccharomyces*, *Candida*), among others, in addition to some animals and plants [5,6,7,8,9]. They can be released extracellularly or be located on cellular cell surfaces. The type and output of the produced biosurfactant are usually species-specific and depend on the substrate used for the microbial growth and their environmental production conditions [10]. They are produced primarily as secondary metabolites and can play important roles in the localization, survival and growth of the producing microorganisms, such as motility, metabolism, attachment/detachment to surfaces, defence from microbial antagonists, biofilm production and backdown and resistance to toxic compounds [11,12].

These surface-active mixture molecules are composed structurally by a hydrophobic tail (unsaturated or saturated fatty acid hydrocarbon chains) and a hydrophilic head (peptide cations, anions or mono-, di-, or polysaccharides), and can form micelles (Figure 1).

For this amphipathic nature and unique structural arrangement, biosurfactants display various physicochemical properties such as excellent surface activity, effective critical micelle concentration and the ability to form microemulsions, humectant, foaming and cleaning activity (Figure 2) [13,14].

In addition, several biosurfactants have other interesting biological properties such as strong antibacterial, antifungal and antiviral activity as well as antioxidant, anticancer and immunomodulatory activity [12,15,16].

For these reasons, in the last two decades, microbial surfactants have been extensively studied, industrially produced and, thanks to their various commercial attractions, applied in many fields worldwide, ranging from paint to petroleum, detergents to water treatment, and food to pharmaceutics, cosmetics and biomedicine [10,15,17,18].

Unfortunately, their large-scale production currently is disadvantageous and is mainly constrained by the high costs related to their downstream processes (over 60–80% of the total costs), such as a long and difficult separation and purification steps that are required for some specific applications (e.g., pharmaceutical) [19,20,21,22]. However, it is believed that the implementation of some alternative strategies such as the selection of new microbial producers of BSs from renewable substrates, such as waste from agro-food processing and the dairy industry, from vegetable oils and animal fat, together with an improvement in fermentation, extraction and purification techniques, can increase their commercial potential [22,23,24].

In 2022, the biosurfactant global market reached USD 1.2 billion. This is forecast to have a compound annual growth rate (CAGR) of 11.2%, increasing to USD 1.9 billion by 2027 [25].

This increase reflects the general attitude worldwide in encouraging the use of nonharmful and eco-friendly products. In fact, for their potential benefits, in recent years, biosurfactants have boomed as an alternative for the chemically synthesised surfactant. Compared to chemical surfactants, biosurfactants have several interesting features and advantages which include biodegradability, high specificity, low toxicity, tolerance to extreme conditions, higher efficiency and an environmentally friendly nature [17,24,26]. In addition, their chemical composition can be modified through genetic engineering manipulation as well as metabolic end-product alterations of the produced strains using biotechnological and biochemical techniques to tailor them to the specific functional requirements [27,28].

However, despite these positive attributes, biosurfactants can only truly be deemed sustainable when the entire production chain, from the fermentation process to the extraction and recovery phase of biosurfactants, is carried out with sustainability in mind. The disadvantages include elevated production expenses and the necessity for purification, especially in specialized applications like pharmaceuticals [12]. The biotechnological procedures employed in biosurfactant synthesis tend to be costly, and the purification of these surfactants present some challenges. Multiple research teams are actively seeking ways to lower the production costs of biosurfactants and increase their sustainability through harnessing readily accessible and renewable bioresources and applying extraction processes that minimize the use of water and organic solvents [12,19]. Another limitation arises from using biosurfactants obtained from pathogenic or potentially pathogenic microorganisms, leading to safety concerns due to the possibility of these microorganisms retaining harmful traits or inducing adverse effects when introduced into some applications.

Biosurfactants are typically classified based on their microbial origin and chemical structure (Figure 3) [29].

They are mainly categorised into two major classes: low-molecular-weight molecules, which comprise lipopeptides, glycolipids, fatty acids and high-molecular-weight molecules, which comprise polymers and particulate materials [10,11,12].

A list of the most prominent biosurfactants and their respective microorganism sources are presented in Table 1. Among them, the low-molecular-weight biosurfactants lower both surface and interfacial tensions at the surfaces, whereas the latter are typically highly efficient in stabilising emulsions [14,30].

Lipopeptides and glycolipids represent the most promising types of biosurfactants for pharmaceutical, therapeutic and biotechnological industries [31,32,33].

Numerous reviews and books have been dedicated to the production, characterization and application of biosurfactants in diverse fields [34,35,36,37]. However, special attention should be paid to the potential and real use of these molecules in the biomedical and pharmaceutical fields.

The main objective of this review is to present a comprehensive update on the latest advancements, research findings and insights into how biosurfactants can be utilized in the pharmaceutical and biomedical fields, building upon our previous review published in 2021 [12]. By examining the recent literature, primarily within the past couple of years, this review aims to shed light on the current state of knowledge and identify emerging trends or promising directions in the application of biosurfactants in the pharmaceutical and biomedical sectors. Compared to our previous literature output [12], particular attention will be directed towards the latest and cutting-edge applications of biosurfactants, with a particular focus on areas such as innovative coatings, drug and gene delivery, wound healing and anticancer. In addition, emphasis will be given to marine organisms and probiotic lactic acid bacteria as natural sources with the potential to provide a wealth of distinctive bioactive compounds such as biosurfactants endowed with antimicrobial properties. Finally, a section detailing the most recent patents that integrate biosurfactants for use in the biomedical and pharmaceutical sectors has been included. Through examining the most recent advancements in the field, this review will shed light on the innovative uses of biosurfactants and their potential in improving the efficacy of drugs and conventional treatments.

**Table 1 pharmaceutics-15-02156-t001:** List of the most common low- and high-molecular-weight biosurfactants and their corresponding microorganisms (adapted from [5,38,39]).

Biosurfactants	Molecules	Producing Microorganisms
Low molecular weight		
Lipopeptides	surfactin	*Bacillus subtilis*
	fengycin, iturin	*Bacillus subtilis*
	pumilacidin	*Bacillus pumilus*
	lychenisin	*Bacillus lycheniformis*
	serrawettin	*Serratia marcescens*
	viscosin	*Pseudomonas fluorescens*
	arthrofactin	*Arthrobacter* sp.
	polymyxins	*Paenibacillus* sp.
	echinocandins	*Aspergillus* sp.
	daptomycin	*Streptomyces* sp.
Glycolipids	rhamnolipids	*Pseudomonas aeruginosa*
	trehalolipids	*Rhodococcus* sp.
	sophorolipids	*Wickerhamiella domercqiae, Starmerella bombicola Trichosporon asahii*
	cellobiose lipids	*Sympodiomycopsis paphiopedili, Cryptococcus humicola, Pseudozyma* sp.
	mannosylerythritol lipids	*Candida antarctica, Ustilago* sp., *Pseudozyma* sp.
High molecular weight	emulsan	*Arthrobacter calcoaceticus*
	mannan lipid protein	*Candida tropicalis*
	liposan	*Candida lipolytica*

## 2. Biosurfactants: A Real Prospect for Biomedical and Pharmaceutical Use?

The discovery and selection of bioactive molecules is one of the main challenges of the modern era in the fight against many diseases and infections. Biosurfactants are very valuable and important products for possible biomedical and pharmaceutical applications (Figure 4); lipopeptides and glycolipids are especially of particular interest. They are characterised by antibacterial, antifungal and antiviral activities, in addition to anticancer, immunological and neurological properties. They are also able to increase in the electrical conductance of bimolecular lipid membranes and the inhibition of fibrin clot formation. Moreover, with their ability to decrease surface tensions between immiscible or miscible liquids, blocking hydrogen bonding and augmenting hydrophilic/hydrophobic interactions, they can be used as antiadhesive/antibiofilm agents on medical devices and applications in transplantation [36,40]. Furthermore, BSs are not only applied to enhance some physical–chemical properties of some pharmaceutical formulations but also to improve the efficacy and performance of these pharmaceutical products. In synergy with nanotechnology, BSs also play essential roles in the development of micro-/nano-based drug delivery systems, self-emulsifying drug delivery systems (SEDDS) and liposomes, controlling the particle size and the stability or solubility of drugs in liquid, semi-solid and solid formulations [41].

It should be remembered that some biosurfactants can be produced by pathogenic microorganisms such as *Pseudomonads aeruginosa* and their biosynthesis is related to the potential pathogenicity of the producing strains and, for these reasons, precautions are essential to avoid allergic reactions and potential skin irritations [34].

Finally, despite their extensive range of applications, it is essential to thoroughly assess the safety of each biosurfactant before considering using it within clinical applications. A limited number of studies conducted on animal models to investigate BSs’ in vivo toxicity and biological activities however highlighted significant challenges in utilising them as therapeutic agents [42,43].

A recent study by Rana et al. (2021) evaluated the toxicity of polymeric nanoparticles loaded with a novel biosurfactant product from *Candida parapsilosis* in a rodent model using biochemical, haematological and histopathological evaluations. The findings indicated that there were no noteworthy variations in haematological parameters between the control and treated groups and only minor, yet not significant, alterations in biochemical parameters. In addition, no toxicological characteristics were observed in the tissue samples, suggesting that the biosurfactant encapsulated within the PLA-PEG copolymeric nanoparticles presents a secure and suitable platform for potential biomedical applications [43]. Other recent toxicity profile testing for rhamnolipid BSs produced by *Marinobacter* and *Pseudomonas* strains showed no cytotoxicity using the in vitro models of human liver and skin cells and no mutagenic/anti-mutagenic potential or significant antioxidant capabilities [44].

Adu et al. (2023) very recently reported on the safety and lack of cytotoxicity of sophorolipid and rhamnolipid (RL) BSs, which showed insignificant effects on human keratinocyte cell line morphology, viability and the production of pro-inflammatory cytokines and potential for use in skincare applications [45].

## 3. Biosurfactants for Innovative Coatings

It is well known that indwelling medical devices are widely used in the healthcare industry and have significantly contributed to improve the treatment of multiple pathologies and the quality of life of patients. However, they can constitute a favourable substrate for the adhesion and growth of microorganisms as biofilms, which are one of the main causes of healthcare-associated infections [46]. In contact with human body fluids, implants are rapidly covered by a conditioning film (composed of proteins, other organic molecules and ions) that attracts microorganisms. After deposition, microbial cells adhere to surfaces, forming micro-colonies that grow as complex communities embedded within an exopolysaccharide matrix. In their sessile form, microbial cells acquire a typical physiological state, becoming more resistant to the host defence mechanisms and up to 1000-fold less susceptible to various antimicrobial agents [47].

One of the most appealing applications of biosurfactants is their use as coating agents in order to prevent biofilm formation on medical devices. They can be utilised to create thin, uniform films on various surfaces, providing numerous benefits, such as enhanced wetting and reduced surface tension, impairing microbial adhesion (Figure 5). With ongoing research and advancements in biosurfactant production and formulation techniques, surface coating methods (such as physical adsorption, bulk incorporation or covalent grafting) have been proposed and used to attach these molecules on a wide range of materials. Moreover, biosurfactant-based coatings exhibit excellent biocompatibility and biodegradability, making them attractive alternatives to conventional synthetic coatings [48,49,50].

In a recent study by Kannan et al. (2021), liposomes encapsulated with a lipopeptide produced by the human skin bacterium *Paenibacillus thiaminolyticus* and copper oxide nanoparticles were able to strongly interfere with Methicillin-resistant *Staphylococcus aureus* (MRSA) and the *Pseudomonas aeruginosa* biofilm formation on urinary catheters. EL-LP-CuNPs possessed marked antibacterial activity with Minimal Inhibitory Concentrations (MICs) of 105 μg/mL (for MRSA) and 89 μg/mL (for *P. aeruginosa*), stimulated reactive oxygen species (ROS) production and accumulation, decreased carotenoid synthesis and exopolysaccharides secretion as well as an induced profound collapse of the cell arrangement and severe destruction on cell walls [51]. The same year, different types of biosurfactants (rhamnolipids, sophorolipids and lipopeptides) were used as anti-adhesive molecules to counteract the colonisation of medical-grade silicone by a cell combination of *Staphylococcus* spp. and *Candida albicans* cultures. The BS coating films, obtained by physical adsorption, significantly limited microbial adhesion and markedly inhibited the formation of the dual species biofilms in a range from 74% to 95% up to 3 days, while reducing their cell viability, metabolic activity and biomass yield at the same time, thus keeping silicone surfaces free from microbial contamination, as confirmed by SEM observations [52]. In another study by Cheffi et al. (2021), lipopeptides Bios-PHKT produced by a *Halomonas venusta* strain isolated from contaminated seawater showed interesting antiadhesive activity towards Gram-negative and Gram-positive human pathogens due to the electrostatic repulsion between the BS coating films on the polystyrene surfaces and bacterial cells. The inhibition of cell adhesion increased in function of Bios-PHKT concentrations, reaching a near-constant level from 0.5 mg/mL onwards, with a maximum effect against *Escherichia coli* (72.3% reduction in cell biomass) [53].

Very recently, rhamnolipids were employed by Sharaf et al. (2022) for the preparation of biosurfactant-coated iron oxide nanoparticles as an innovative multitarget approach to combat some food-borne *Escherichia coli* serotypes and MRSA. In particular, the antibiofilm and antiadhesive properties of rhamnolipid-coated Fe_3_O_4_ nanoparticles (NPs) were combined with two antimicrobial drugs, gallic acid (GA) and p-coumaric acid (p-CoA). The biosurfactant-coated iron oxide nanoparticles (RHL-Fe_3_O_4_@PVA@p-CoA/G) significantly interfered with growth and significantly limited the biofilm formation when down-regulating operon I*caABCD*, which is responsible for the formation of the slime layer in *S. aureus*, and *CsgBAC*, which is responsible for the production of curli fimbriae *E. coli* [54]. In another study, a glycopeptide biosurfactant produced by *Lactobacillus delbrueckii* alpha2 was applied at different concentrations (from 2.5 mg/mL to 25 mg/mL) on polystyrene surfaces for the evaluation of its antiadhesive potential against four pathogenic strains (*Klebsiella* spp., *Bacillus* spp., *S. aureus*, *E. coli* and *Pseudomonas* spp.) with percentages of inhibition ranging from 56% to 71% at the 25 mg/mL concentration. In addition, the authors demonstrated that a glycoprotein-coating of medical-grade silicone tubes was effective in preventing the growth of *S. aureus* and *E. coli* biofilms [55].

Findose et al. (2023) evaluated the protective effect of rhamnolipid coating against the attachment and formation of biofilms in ESKAPE pathogens such as *Acinetobacter baumannii* and *Enterococcus faecium*. The pre-treatment of 96-well plates with BSs prevented the deposition and adhesion of bacterial cells on surfaces in a concentration-dependent and strain-dependent manner, with the most significant results at a concentration of 31.2 μg/mL for *E. faecium* with a reduction of 91% and at 250 μg/mL for *Ac. baumannii* with a reduction of 76% [56].

Section 6 elucidates a collection of recently patented and ground-breaking methodologies utilised for the functionalization of biomaterials with biosurfactants, thus evidencing the strong scientific interest in these anti-adhesive coatings.

## 4. Biosurfactants as Biological Control Agents

The declining effectiveness of conventional antimicrobials due to the increasing number of multi-resistant pathogens highlights the urgent need for alternative approaches [57,58]. Biosurfactants can effectively affect the growth of a wide range of pathogenic microorganisms, including both Gram-negative and Gram-positive bacteria, as well as various types of fungi. Furthermore, unlike synthetic drugs, these molecules can offer unique antimicrobial mechanisms of action as reducing cell surface hydrophobicity, disrupting membrane integrity, increasing its permeability, altering protein conformation and inhibiting membrane functions (transport and energy generation) or blocking the quorum-sensing system and down-regulating gene expression (Figure 6) are difficult for microorganisms to overcome and develop resistance to, thus making biosurfactants a valuable tool for developing sustainable and environmentally conscious approaches to combating microbial infections [12,59,60].

### 4.1. Biosurfactants as Antibacterial and Antifungal Agents

#### 4.1.1. Marine Microorganisms

The vast and diverse marine ecosystem harbours a plethora of microorganisms that have adapted to survive in extreme conditions and could offer a rich source of unique bioactive compounds; among which, novel biosurfactants with potent antibacterial and antifungal activities can effectively combat microbial infections. In addition, harnessing the biosurfactants produced by marine microorganisms not only opens up possibilities for developing alternative and sustainable antimicrobial agents, but also contributes to the conservation and exploration of marine biodiversity [61,62,63].

Lipopeptides from marine *Bacillus amyloliquefaciens* showed antibacterial activity against *S. aureus* CCM 4223 planktonic cells with MIC = 15 mg/mL, and gradually limited its biofilm formation in a dose-dependent manner up to a complete inhibition at MIC by down-regulating the expression of biofilm-associated genes *fnbA*, *fnbB*, *sortaseA* and *icaADBC* operon [64].

The rhamnolipid mixture GBB12 produced by *Shewanella algae*, a marine isolate from the Persian Gulf, exerted significant antimicrobial activities against Gram-negative and Gram-positive clinical pathogens such as *Streptococcus pneumoniae*, MRSA, *P. aeruginosa*, *E. coli*, *K. pneumoniae* and *Ac. baumannii*, with MIC values ranging between 7.8 mg/mL and 12.5 mg/mL. In addition, the glycolipid biosurfactant was able to totally, or almost completely, inhibit the growth of MRSA and *Ac. baumannii* biofilms, and exhibited considerable dislodging activity on the pathogenic preformed biofilms with a maximum effect (>90%) against *P. aeruginosa* and *Ac. baumannii* [65,66].

More recently, a rhamnolipid mixture from the Antarctic marine bacterium *Pseudomonas gessardii* M15 demonstrated marked bactericidal activity against a panel of *S. aureus* multidrug-resistant clinical isolates with MICs ranging from 12.5 to 50 μg/mL and MBCs equal to MIC or 2 × MIC values. M15RL killed the entire population of *S. aureus* 6538 in 30 min and 5 min at MIC and 2 × MIC, respectively, and as revealed by SEM visualisations, induced the accumulation of intracellular material due to cellular damage and loss of structural integrity. In addition, rhamnolipids were able to significantly compromise *S. aureus* and MRSA biofilms at all the stages of their development and, when applied on cotton swabs, had completely eradicated the bacterial load, radically preventing the proliferation of the pathogens [67].

#### 4.1.2. Probiotic Lactic Acid Bacteria

Lactic acid bacteria (LAB) are typically known for their probiotic properties and benefits on human health. The pharmaceutical and biomedical industries are directing their attention towards the search for LAB that produce cell-bounded and excreted biosurfactants due to their inhibitory activity against the growth of various pathogens, including bacteria, fungi and viruses [68,69,70].

In a recent study, the cell-bound biosurfactant from a *Lactobacillus rhamnosus* strain isolated from human breast milk inhibited various bacterial pathogens’ (such as *P. aeruginosa*, *S. aureus* and *E. coli*) adhesion to surfaces and promoted the eradication of their pre-formed biofilms by altering the integrity and viability of bacterial cells as well as by reducing the total exopolysaccharide matrix content [71]. The glycolipid produced by a *Lactobacillus plantarum* strain isolated from a yoghurt sample decreased virulence in *P. aeruginosa* and *Chromobacterium violaceum*, disabling the quorum-sensing control system. A dose-dependent inhibitory effect of the tested biosurfactant on swarming motility and biofilm formation, violacein, acyl homoserine lactone, pyocyanin and exopolysaccharide production, and a reduction in LasA protease and LasB elastase activities, were reported [72]. In another study, the non-homogeneous lipopeptide produced by *Lactobacillus crispatus* BC1 in co-incubation conditions demonstrated moderate antibiofilm (up to 66%) and dislodging (up to 43%) activity against *Candida albicans* and *Candida* non-*albicans* clinical isolates. The biological effect was then increased through the inclusion of the lipopeptides inside conventional liposomes and was further enhanced by the coverage of the nanocarriers with hyaluronic acid, leading to percentages of reduction up to 85% for biofilm formation and up to 81% for biofilm dispersal [73].

#### 4.1.3. Other BS Producers

The mixture of lactonic and acidic sophorolipids (SLs) produced by *Starmerella riodocensis* GT-SL1R sp. nov. strain displayed good antifungal activity against an opportunistic pathogen, *Candida albicans*. The SLs effectively inhibited hyphal transition in a dose-dependent way, starting from 32 μg/mL up to 500–1000 μg/mL, where the treatment reduced cell survival and the fungal strain remained in yeast formation. Furthermore, SLs at 500 μg/mL showed good potential in preventing biofilm formation and reducing it, both in terms of metabolic activity and biomass, by approximately 50%, leading to fragmented yeast cells with swollen, wrinkled, punctured and diminished fragmented bodies [74].

In a study carried out by Haddaji and co-workers (2022), the lipopeptides extracted from a probiotic *Bacillus* strain were tested in terms of antibacterial and antibiofilm abilities against four vaginal-associated pathogenic *Staphylococcus* strains. The tested BSs inhibited the growth of *Staphylococcus* clinical isolates known to have multidrug resistance, giving inhibitory zones on agar plates with diameters ranging between 27 and 37 mm and cellular growth inhibition at an MIC of 1 mg/mL; this was quite effective against the biofilm formation of the different pathogenic strains tested [75].

In the study conducted by Manikkasundaram et al. (2022), the glycolipid HRB1 was evaluated for its biomedical potential by testing its antifungal activity against *Magnaporthe grisea* and *Alternaria* spp., its antibiofilm activity against *P. aeruginosa* and its ability in blocking *C. violaceum* quorum-sensing signalling; they concluded that the glycolipid had antiphytofungal, antibiofilm, anti-quorum-sensing, antioxidant, anticancer and dye-degradation abilities [76].

### 4.2. Biosurfactants as Antivirals

To date, viral infections are one of the most remaining challenges that the scientific community faces daily. Antiviral drug resistance is mainly developed by the amount of viral antigenic peptides that are inactive or not properly anchored for maximum efficacy [77]. In general terms, the genetic materials of viruses are enclosed by protein layers known as capsids, whereas in virions, the capsids are surrounded by lipid bilayers that contain viral proteins which facilitate binding to the host cells [78]. Given the amphiphilic properties of biosurfactants, it has been shown that they are able to mediate the interaction with the hydrophobic domain within the lipid membrane of enveloped viruses, promoting disruption [79]. Following this context, several (bio)surfactant delivery systems for different applications have been reported, including micro-/nano-based drug carriers, microspheres, micro-/nanoemulsions, liposomes, solid lipid nanoparticles (SLNs), self-emulsifying drug delivery systems (SEDDS), novel powders, hydrogels and polymeric micelles [80,81,82,83,84,85].

Cirrhosis is a severe scarring of the liver, where in many cases, is due to the presence of Hepatitis C Virus (HCV). Different treatments based on interferon-free antivirals are the current methods to treat HCV infections; however, the number of successful treatments equals the number of patients that have acquired HCV due to the constitution of the virus itself [78]. HCV contains a positive single-strand RNA that is associated with core protein and encased by two glycoproteins E1 and E2 [78], of which are responsible for the viral entry, while the most important nucleoid-associated protein for replication is the RNA-dependent RNA polymerase (NS5B) [86]. Hegazy et al. (2021) studied the antiviral properties of a biosurfactant produced by haloarchaeon *Natrialba* sp. against HCV, specifically against E2 (binding receptor) and NS5B (RNA-dependent RNA polymerase), and for the first time, a biosurfactant has shown antiviral properties against HCV in dual antiviral mode: through reactivity against HCV E2 and as an inhibitor of NS5B and HSV polymerase, showing an advantage in comparison to the single antiviral mode [78].

One Health is a well-known collaborative and multidisciplinary approach that works at different global levels with the aim of addressing the existing connection between people, animals, plants and their environment. Following this connection, animal viral diseases are as important as any other for the impact that they have towards people, plants and the surrounding environment. Newcastle disease virus (NDV) is a poultry- and bird-based contagious viral disease affecting the nervous, respiratory and digestive systems. Behzadnia et al. (2022) have reported the first study showing the effect of two *Lactobacillus*-derived biosurfactants (waste-based and synthetic) against NDV LaSota strains, where the biosurfactant products inactivated the NDV LaSota strain at concentrations of 3.75 mg/mL and 7.5 mg/mL; this is much lower than the lipopeptide biosurfactants synthesized from *B. cereus*, which had an inhibitory concentration of 10 mg/mL [87]. Despite having great antiviral properties, biosurfactants have not been used in any clinical settings; they have, however, displayed their potential applications in the therapy and pharmacology sectors.

The recent outbreaks have emerged, without a doubt, as the world’s state of flow has changed. Despite all efforts and the great advances and accomplishments that the medicine and research field have made, to date, the diagnosis and control of viral outbreaks is still a challenge due to the high mutation rate that viruses have. The last report published in May 2021 showed that around 170 million people were affected by COVID-19 [88]; a total of 3.5 million people died worldwide, and the world’s healthcare systems, economic circumstances and the social wellbeing of people were hugely impacted.

As mentioned before, biosurfactants have been reported as antiviral agents due to their physical properties, altering viral membrane structures and disrupting their outer covering, finally causing eradication [89]. Glycolipids are one of the groups of biosurfactants more widely studied. Sophorolipids, a group of glycolipids produced by *Starmerella bombicola*, have shown properties as antimicrobials, biofilm disruptors, immunomodulators and anti-inflammatory agents. It has been previously reported that via the acetylation of the sophorose head groups, they have been active against both the Herpes virus and HIV virus. This modification is considered to promote its antiviral and cytokine-stimulating properties, increasing the hydrophobic side of the molecule [42,90,91]. Given their unique characteristics, biosurfactants can make a difference to the future of the healthcare system as they could be considered as an alternative natural antiviral agent and may be effective against many other viruses in the future [92].

## 5. Perspectives of Biosurfactants for Wound-Healing, Anticancer and Immuno-Modulatory Applications

Potential contributions of biosurfactants to improving human health and well-being have also been described, in recent years, in the context of wound healing, where they have shown potential in promoting tissue regeneration and accelerating the healing process. Additionally, their antimicrobial properties make them attractive candidates for combating infections in wounds. Moreover, biosurfactants have exhibited anticancer activities by inhibiting tumour growth and inducing apoptosis in cancer cells. Furthermore, they possess immuno-modulatory properties, making them potential candidates for immunotherapy and immunomodulation strategies [12,40,93,94].

### 5.1. Wound Healing

Wound healing is a natural biological process where four distinctive phases are involved: haemostasis, inflammatory, proliferation and remodelling [95]. Although wound healing has been driven by an innate immune response [96], multiple factors can lead to impaired wound healing; for example, a chronic wound generally takes over 6 weeks to heal, and can lead to major hurdles regarding therapeutic approaches, since any therapy must effectively be sequenced to the appropriate stage [94]. Therefore, biocompatible and safety approaches are ideal characteristics for topical wound therapies [97].

The effect of biosurfactants on wound healing has been a point of interest for over 15 years due to their therapeutics properties [97]. Cheffi et al. [53] evaluated the effect of Bios-PHKT (a lipopeptide molecule) on HEK-293 cells, showing that Bios-PHKT-stimulated cell migration and proliferation can be compared to the controls. Interestingly, the concentrations required of the Bios-PHKT to achieve wound healing were lower than that exhibiting cytotoxic effects on the HEK-293 cells. Lipopeptides are one of the biosurfactants more widely studied in terms of their emulsion, antitumour and antimicrobial properties, and their effect on the immune response [98]. Ohadi et al. [99] reported that a lipopeptide generated by *A. junii* B6 helps to protect mice cells from free radicals’ damage with evidence of recovery of the cells, using an excision wound model.

Li et al. (2007) [100] highlighted the importance of using angiogenesis in wound repair, making emphasis on growth factors and the role that those factors play on the extracellular matrix. Following that context, Afsharipour et al. (2021) [101] assessed the impact of lipopeptides (LPBs) on angiogenesis. They observed that it triggered tube formation and promoted the migration of endothelial cells, marking significant progression in the angiogenesis process. Additionally, LPB exhibited notable enhancements in the protein expressions of HIF-1α and VEGF in Human Umbilical Vein Endothelial Cells, as compared to the control group. This is in line with the effect of another lipopeptide, surfactin (a biosurfactant produce by *Bacillus subtilis*), which displays wound-healing activity most likely through inducing keratinocyte migration and enhancing the expression of VEGE and HIF-1α proteins [102]. More recent studies displayed that it is not just the amphiphilic properties of LPB which play an important role on wound healing, but the size of the molecule within effective formulations in in vivo models. Afsharipour et al. (2021) [101] have shown that it is possible to use a nano-lipopeptide biosurfactant (NLPB) formulation from LPB, highlighting the maximum efficacy at smaller sizes of LPBs. Lipopeptides are not the only type of biosurfactant that displays a positive effect on the treatment of wound healing. Sophorolipids have been well known for having strong antimicrobial properties [103] which contribute to their significant healing activity on wounds when compared to commercial creams [104]; this aligns with results previously reported by Sekhon Randhawa and Rahman [105], where there was a notable effectiveness of rhamnolipids on different skin treatments. At present, the commercial skincare and cosmetic products that contain biosurfactants in their formulation include Sopholiance^TM^ S (face cleaner, deodorant and shower gel), Relipidium^TM^ (face and body moisturizer), and Kanebo skincare (UV filter, moisturizer and cleanser) [106,107]. The excellent healing activity of biosurfactants makes them a potential alternative in cosmetic formulations through the replacement of their chemical counterparts with the purpose to produce more environmentally friendly products.

### 5.2. Anticancer Agents

Despite significant advances in cancer therapy, cancer remains the second leading cause of death worldwide. Specifically, the World Health Organization (WHO) presented, in 2020, data which estimated 10 million deaths, with new cases rising to 19.3 million [108]. To combat this life-threatening disease, new cancer treatments have been constantly developed over the years, with radiotherapy and chemotherapy remaining the main options in cancer treatment [109,110]. Despite some positive aspects of these therapies, the increasing mortality rates can be attributed to the lack of specificity of anticancer drugs for cancer cells, leading to severe side effects, low success rates [111], and the development of multidrug resistance by cancer cells [112]. Given the current circumstances, there is an urgent need to develop highly targeted and less toxic molecules for effective cancer therapy. Therefore, efforts have been made worldwide to find new anticancer agents and drugs that can selectively target and sensitize cancer cells [113]. Natural anticancer drugs should be explored as a replacement for chemical drugs to overcome their limitations. Microorganisms, particularly bacteria, have gained significant attention as a potential source for new anti-cancer compounds due to their high biodegradability, specificity and low toxicity [114]. Furthermore, biosurfactants have recently emerged as promising alternative molecules for treating various types of cancer, including pancreatic, breast, cervical, oral, colon, lung and liver cancers [114,115]. They have demonstrated potential for the treatment of cancer as they are able to regulate certain functions in mammalian cells so that they can prevent the abnormal progression of cancer, resulting in the inhibition of cell proliferation, viability and migration [116]. A study conducted by Adu SA et al. showed that certain biosurfactants, including glycolipids and lipopeptides, have the ability to inhibit tumour cell proliferation and survival [117].

Moreover, in this study, the authors demonstrated that glycolipids, especially sophorolipids and rhamnolipids, have different effects on human skin cells depending on their chemical structure. In particular, lactic mono-rhamnolipids and sophorolipids were found to have a significant cytotoxic effect on malignant melanoma cells (SK-MEL-28) compared to healthy human keratinocytes (HaCaT). In addition, the study found that glycolipids induced cell death in melanoma cells by necrosis, and sophorolipids significantly inhibited the migration of SK-MEL-28 melanoma cells, indicating their potential as antimetastatic agents. This study suggests that glycolipids are potential candidates for novel therapies against skin cancer and could be used to replace synthetic surfactants in sunscreens. Further research using appropriate models is needed to fully understand the mechanisms and explore the potential of glycolipids as targeted anticancer agents for malignant melanoma [117]. In a recent study, Haque et al. (2021) investigated the mechanism of action of glycolipids, specifically acidic and lactonic sophorolipids, bolalipids and glucolipids against cancer cells. The experiments were performed with three different cell lines: the lung cancer cell line (A549), the breast cancer cell line (MDA-MB 231), and the mouse skin melanoma cell line (B16F10). The results suggest that glucolipids inhibit tumour cell migration, possibly through interference with actin filaments, and that both lactonic sophorolipids and glucolipids induce the formation of reactive oxygen species in cells. In addition, these biosurfactants altered the mitochondrial membrane potential and eventually resulted in cell death by necrosis. [118]. A lipopeptide (LP) produced by *Bacillus halotolerans* has also shown promising anticancer activity against the MCF-7 human breast cancer cell line in very low concentrations. In this case, the IC50 values of the purified LP on MCF-7 cells were 46.1 μg/mL after 24 h, 42.16 μg/mL after 48 h, and 40.4 μg/mL after 72 h of incubation. These values indicate that the purified LP was effective in the growth inhibition of MCF-7 cells in a dose-dependent manner. It is noteworthy that the purified LP did not show cytotoxic effects on normal cells (HEK-293), specifically normal human embryonic kidney cells. Moreover, after 24 h of treatment with a concentration of 45 μg/mL, the MCF-7 cancer cells underwent apoptosis, as evidenced by the flow cytometry analysis [119]. In addition to their high potential as anticancer drugs, biosurfactants can also be used as carriers or delivery systems for anticancer drugs (see Section 6) [120].

### 5.3. Immuno-Modulatory Agents

Biosurfactants have demonstrated immunomodulatory activity, making them valuable agents in the field of immunology. These natural compounds can act as ligands, binding to immune cells and influencing their activation and function. They have been shown to affect different types of immune cells such as macrophages, neutrophils, B cells and T cells, leading to the production of cytokines and chemokines so that an effective adaptive immune response can be generated [121]. For example, as reviewed by Thakur et al. (2021), rhamnolipids can act as immunomodulators and regulate the humoral and cellular immune response, leading to the release of pro-inflammatory cytokines [122]. In addition, biosurfactants are able to modulate the balance between proinflammatory and anti-inflammatory factors and thus have a regulatory effect on immunological diseases and disorders. Recently, Kwak et al. (2022) investigated the therapeutic effect of dietary supplementation with sophorolipids in a mouse model of colitis induced by DSS (dextran sodium sulphate). The results suggest that dietary supplementation with SLs has beneficial effects on gut health; for example, it reduces inflammation, upregulates the gene expression of protective factors (MUC2, IL-10 and TGF-β) and improves mucosal barrier function. Thus, it has potential as a therapeutic intervention for colitis [123]. Moreover, as reported by Daverey et al. (2021), the purified natural mixtures of sophorolipids have exhibited immunomodulatory properties by reducing inflammatory cytokines like IL-1β and TNF-α and increasing anti-inflammatory cytokines (TGF-β1) in animal models [124]. Due to their natural origin, biocompatibility and diverse biological activities, biosurfactants hold great potential for the development of immunotherapies, vaccine adjuvants and other therapeutic applications in the field of immunology. Adu SA et al. (2023) investigated the effects of highly purified glycolipids, including acidic and lactonic sophorolipids (SLs) and mono-RL and di-RL congeners, on human keratinocytes (HaCaT cells) compared to the commonly used surfactant SLES (sodium lauryl ether sulphate). They observed that the glycolipids had varying effects on HaCaT cells depending on their chemical structure. Acidic SL and mono-RL had minimal impact on cell morphology, viability and pro-inflammatory cytokine production compared to SLES. Notably, di-RL significantly reduced IL-8 production and CXCL8 expression, while increasing IL-1RA production and IL1RN expression in cells stimulated with lipopolysaccharides (LPSs). These effects were not observed with SLES or other glycolipids. These findings suggest that glycolipids could serve as potential alternatives to synthetic surfactants in skincare formulations and may have immunopharmacological implications for skin infections like psoriasis [45]. In another in vitro study, conducted by Sharifi et al. (2023), researchers had investigated the effects of a lipopeptide biosurfactant (LPB) produced by *Acinetobacter junii* B6 on *Leishmania tropica* infection and cytokine gene expression in infected macrophages alone and in combination with glucantime^®^ (meglumine antimoniate, MA). When LPB and MA were combined, there was a substantial reduction in the expression of Th2 cytokines and a significant increase in Th1 transcription factors and cytokines compared to individual treatments. This suggests that LPB + MA could be a potential therapy for anthroponotic cutaneous leishmaniasis, opening possibilities for novel treatments against leishmaniasis and other protozoan parasites [125].

Developing efficient formulations combined with safe and effective adjuvants remains a significant challenge in vaccine development. As extensively reviewed by Khodavirdipour et al. (2022) and Kumari et al. (2023), lipopeptides derived from *Bacillus subtilis* have been identified as non-pyrogenic, non-toxic and effective immunological adjuvants for antigenic priming and vaccine design. These lipopeptide adjuvants activate the immune system through Toll-like receptor (TLR2) signalling, and by recognizing viral peptides coupled with major histocompatibility complex (MHC) class 1, lipopeptides can stimulate cytotoxic T lymphocytes that are specific to the virus. They can enhance immunity when combined with other medications or serve as an effective solution when primary immunity against a virus is lacking. For example, surfactin possesses emulsifying properties and can serve as an immunological adjuvant in vaccines or drugs [89,126].

## 6. Utilizing Biosurfactants as Adjuvants in Medicine: Drug Delivery Systems

Drug delivery systems play a crucial role in pharmaceutical and medicinal sciences, serving as formulations or devices/vehicles that enable the controlled administration and release of the active ingredients to specific parts of the body via various routes depending on the desired effect and the nature of the disease, thereby enhancing efficiency and safety [38]. These numerous advantages offered by these systems include optimal drug loading capacity without loss of the drug, aqueous solubility and improvement in bioavailability as well as facilitated and controlled transport of the active substance across membranes to the intended site, maximizing efficacy [127]. Nanoparticles, nanoemulsions, microemulsions and liposomes are distinct types of drug delivery systems utilized in various medical applications. Nanoparticles are dispersed particulates or solid particles that have a size range of 10–1000 nm. They can be prepared as nanospheres or nanocapsules. Nanoparticles have been used in various treatment strategies, including drug delivery, due to their unique physical properties, their potential use in controlled release and their ability to protect drugs or other molecules with biological activity against their neighbouring environments, thus improving their bioavailability and therapeutic index [128].

Nanoemulsions and microemulsions are therefore colloidal systems composed of oil, water and surfactants. They have small droplet sizes and can enhance drug solubility and stability [129]. Liposomes are vesicles formed by hydrating a mixture of cholesterol and phospholipids. They improve drug efficacy by delaying clearance from the circulation system and protecting the drug from their biological environment [130].

Biosurfactants can be employed in drug delivery due to their unique surface-active properties and benefits. Overall, the use of biosurfactants in drug delivery holds promise for improving drug solubility, achieving controlled release, ensuring biocompatibility, providing protection and enabling targeted delivery. These advantages make biosurfactants attractive candidates for developing innovative and efficient drug delivery systems.

An investigation on the effect of two biosurfactants (BSs) produced by *Lactobacillus gasseri* BC9 and *Lactobacillus crispatus* BC1 and on the skin permeation profile of hydrocortisone (HC) was carried out. Both BSs increased the solubility of HC, particularly at concentrations above their critical micellar concentrations (CMCs). At concentrations below the CMC, drug permeation through the skin was improved due to the formation of a superficial lipophilic environment and the interaction between BS and the stratum corneum (SC). When skin hydration tests and differential scanning calorimetry (DSC) analyses were carried out to further investigate the BS interaction with the outermost layer of the skin, the data showed that both BS products may be considered as new promising green excipients for drug permeation enhancement through the skin [131]. Another study by the same research group explored the potential of the *L. gasseri* BC9 biosurfactant as a natural excipient to enhance the hydrocortisone solubility and nasal delivery and its mucoadhesive properties at concentrations both below and above the CMC. Diffusion studies performed using sheep nasal mucosa with BC9-BS at a concentration below the CMC indicated that BC9-BS interacted with the nasal mucosa components, leading to increased drug solubilization and permeation at lower concentrations, suggesting that BC9-BS has potential as a promising alternative to chemical surfactants for nasal drug delivery applications [132]. Lipopeptides produced by *Bacillus velezensis* showed antibacterial activity on the surface of both vegetative cells and spores. Strains with the highest levels of activity also exhibited detectable lipopeptide micelles, which were heat- and gastric-stable and able to entrap other antimicrobials produced by the host bacterium itself. In addition, the naturally produced micelle formulations were able to entrap other antimicrobial compounds, such as vancomycin, resveratrol and clindamycin, and the incorporation of antibiotics into micelles increased their activity, suggesting their potential use to enhance drug delivery [133]. In a study by Lassenberger et al. (2021), silk-based composite hydrogels incorporating anionic biosurfactant assemblies (sophorolipids SL-C18:0 and SL-C18:1) were generated to enhance the properties of silk fibroin. The inclusion of sophorolipid assemblies accelerated the gelation of silk fibroin, suggesting promising potential for improving the functionality and mechanical properties of SF-based hydrogels. These advancements present exciting opportunities for controlled cell culture applications, tissue engineering and drug delivery [134].

Interestingly, sophorolipids have also been used as green delivery carriers to create scalable, cost-effective and environmentally friendly nanopesticide systems for agronomic applications [135].

### 6.1. Nanoparticles

Biosurfactants have emerged as highly promising candidates for the development and synthesis of eco-friendly bioactive nanoparticles, thereby replacing synthetic surfactants. Numerous studies in recent years have described the use of biosurfactants as substitutes for conventional surfactants in the field of nanoparticle synthesis, exhibiting significant promise for applications in biomedical science, including antimicrobial activity, drug delivery, controlled release and anticancer activity [136,137,138,139,140,141].

#### 6.1.1. Nanoparticles with Antibacterial Activity

The utilisation of biosurfactants for the production of nanoparticles with antibacterial activity offers a promising avenue in nanotechnology and biomedical applications. On the one hand, biosurfactant-based nanoparticles have demonstrated great potential in combating bacterial infections as they can efficiently target and destroy bacterial cells through multiple mechanisms. Biosurfactants can act as stabilisers, reducing agents and templates in the synthesis of nanoparticles, thus enabling precise control over their size, shape and surface properties. On the other hand, the internalisation into the metal nanoparticles contributes to a further increase in the antimicrobial activity of BSs, enhancing their biodistribution and limiting unfavourable interactions with unspecific targets. For example, in Chauhan et al. (2022), lipopeptide OXDC12 silver nanoparticles possessed remarkable antibacterial activities against Gram-negative *Salmonella typhimurium*, *Klebsiella pneumnoniae* and *E. coli* strains, with MIC values lower than the lipopeptide mixture alone [142]. In 2020, Shikha and collaborators developed gold nanoparticle (AuNps) green synthesis using sophorolipids (SLs) as a non-toxic reducing and stabilising agent; they demonstrated that the obtained AuNPs-SL had higher efficacy in inhibiting the metabolic activity of viable Gram-negative *Vibrio cholerae* and *E. coli* cells as well as in killing *V. cholerae* in its non-multiplying stage, compared to AuNPs or SL alone. In addition, the authors showed that the bactericidal activity of AuNPs-SL on *V. cholerae* cells was attributable to inhibiting the the dehydrogenases of the respiratory chain or inducing morphological changes that lead to cell membrane disruption and intracellular fluid leakage [143]. In another study, rhamnolipids were applied to synthesise CuO nanoparticles to counteract multi-drug-resistant pathogens. RL-CuO NPs showed excellent antibacterial activity both against Gram-positive strains, such as *Streptococcus mutans*, *S. aureus* and *Enterococcus faecalis*, and Gram-negative strains, such as *Shigella dysentriae* and *Salmonella typhi*, with MIC values at concentrations around 7.8 µg/mL and 250 µg/mL, respectively, demonstrating that RL and CuO NPs can combine their activities in synergy to increase cell membrane permeability and promote bacterial cell inactivation [144].

In a recent study by Falakaflaki et al. (2022), a cryogel nanocomposite scaffold enriched with the antimicrobial agent usnic acid encapsulated in rhamnolipid (RH) biosurfactant nanoparticles were designed with dual purpose for antimicrobial activity against *Staphylococcus aureus* biofilm and bone regeneration. Rhamnolipid biosurfactant in the preparation of nanovesicles improved usnic acid solubility and enhanced the antimicrobial effects of the drug. In particular, the biocompatible usnic acid/rhamnolipid containing cryogel scaffolds had an osteogenic effect, and an increase in the expression of bone repair markers was detected by RT-PCR. Strong antibacterial effects of the scaffold on *S. aureus*, with an inhibition zone of 1.4 ± 0.2 cm and antibiofilm activity of 43.7 ± 1.6%, were demonstrated [145].

#### 6.1.2. Nanoparticles for Drug Delivery

Numerous investigations have been carried out on the application of rhamnolipids or lipopeptides in nanoparticles’ stabilisation for their use in the drug delivery field, with particular interest toward cancer therapy and transdermal administration.

For example, Müller et al. (2017) used several rhamnolipids as nano-carriers for different hydrophobic drugs like dexamethasone, Nile red, or tacrolimus for skin delivery in ex vivo studies, and demonstrated that rhamnolipid nanoparticles efficiently deliver Nile red into the skin without causing toxic effects at concentrations higher than CMC values [146]. In a paper by Lewińska et al. (2022), poly(D,L-lactide) nanoparticles stabilised by surfactin were developed for potential administration transdermally. The nanoprecipitation approach was used to obtain nanoparticles. Skin permeability tests obtained on pig ear skin proved the enhanced ability of nanoparticles to penetrate deeper into the epidermis, demonstrating the suitability of surfactin-stabilised poly(D,L-lactide) nanoparticles as biocompatible options for transdermal applications [147].

As for cancer therapy applications, in 2019, rhamnolipid nanoparticles loaded with hydrophobic photosensitizers were injected into the SCC7-cancer-bearing mice. Interestingly, nanoparticles promoted both a significant accumulation of “pheophorbide a” into the targeted tissue and cancer suppression by photodynamic therapy [148]. In 2020, the lipopeptide from *Acinetobacter junii* B6 was used to produce gold nanoparticles; the cytotoxic activity against U87, A549 and MCF7 cancer cell lines was dose-dependent with IC50 values of 89.08 ± 0.4 μg/mL, 646.12 ± 0.5 μg/mL and 3.37 ± 0.1 μg/mL [149]. Another study explored the possibility to use rhamnolipids for the preparation of double-emulsion nanoparticles (NPs) containing doxorubicin and erlotinib (RL-NP-DOX-ER) as a drug delivery system for combination therapy to provide an efficient drug delivery to tumour tissues with a synergistic effect. The double-emulsion method enabled the simultaneous loading of hydrophilic doxorubicin and hydrophobic erlotinib in the NPs, and biosurfactants provided stable surface coating. The resulting nanoparticles demonstrated a rapid cellular uptake and the synergistic killing of tumour cells. Notably, RL-NP-DOX-ERL exhibited enhanced tumour suppression compared to the control groups treated with free drugs or nanoparticles containing a single drug, highlighting the potential of double-emulsion nanoparticles and rhamnolipid coating for efficient tumour combination therapy [150].

In the work proposed by Wadhawan et al. (2022), a novel biosurfactant isolated from *C. parapsilosis* loaded into polymeric nanoparticles was investigated as a promising therapeutic system against MDA-MB-231 breast cancer cells. The nanoparticles were prepared using a polymer material called PLA-PEG (polylactic acid–polyethylene glycol), which contained BS concentrations ranging from 1.25 to 20 μg/mL; these nanoparticles were compared with pure nanoparticles without biosurfactant and nanoparticles conjugated with folic acid (FA) to target cancer cells. The formulation with folic acid showed maximal internalization and superior cytotoxicity compared to non-targeted formulations against MDA-MB-231 cells. In addition, this formulation was observed to induce apoptosis in the breast cancer cell line, thereby killing the cancer cells. It is important to note that these results were specific to the MDA-MB-231 breast cancer cell line used in the study, and further studies would be required to determine the cytotoxic and apoptotic effects of the biosurfactant-loaded nanoparticles in other cancer cell lines or in vivo models [151].

### 6.2. Microemulsions and Nanoemulsions

Micro- and nanoemulsions are dispersed systems extensively utilized for effective and precise drug delivery via different administration routes. Interestingly, the disparity between micro- and nanoemulsions is not solely determined by their size scale, as the names imply, as both systems can contain droplets with diameters below 100 nm. The primary distinction between these systems lies in the method of achieving the droplet size: nanoemulsions involve a mechanical reduction process, whereas microemulsions form spontaneously [152]. A microemulsion delivery system based on commercial biosurfactant sophorolipids (SLs) was developed to improve the solubility and stability of Xanthohumol, a compound with numerous physiological activities such as antioxidant, antimicrobial, anticancer, anti-inflammatory, anti-osteoporosis and neuroprotective activities. The SL-based microemulsion system increased the solubility of Xanthohumol by about 4000 times and extended its half-life to over 150 days, making it a potential green solubilization and delivery method for Xanthohumol and other hydrophobic drugs [153].

In a study published in 2022, submicron emulsions (ESEs) loaded with etoposide, a traditional anticancer chemotherapeutic agent, were prepared using lactonic sophorolipids (LSLs) and acidic sophorolipids (ASLs), compared to the chemical surfactant Tween-80. The results showed that ASL exhibited superior properties and activities compared to LSL and Tween-80 in the ESE formation. The ASL-ESE demonstrated a higher drug-loading capacity and slower drug release rate. It also significantly increased antitumour activity against the ovarian cancer cell line A2780 through apoptosis when compared to Tween-ESE and commercial etoposide injections. Additionally, ASL-ESE showed no haemolysis and displayed comparable long-term and autoclaving stability to Tween-ESE; this therefore highlights the exceptional capabilities of ASL in the ESE formation, efficacy enhancement and improvement in safety [154].

Stable oil-in-water nanoemulsions were prepared by combining different biosurfactants (plant-derived saponin and microbial rhamnolipid and surfactin) using high-energy ultrasonication techniques. The rhamnolipid–surfactin system formed a more stable nanoemulsion compared to the combinations rhamnolipid–saponin and surfactin–saponin. The study suggested that the optimal synergy between mixed biosurfactants at the oil–water interface leading to stable nanoemulsion is primarily dictated by the type and composition of the biosurfactants used in the formulation. In addition, antimicrobial and scavenging investigations of stable nanoemulsions revealed that two biosurfactant systems showed comparable efficacies to single biosurfactant nanoemulsions [155].

In another work, rhamnolipids were used in combination with tea-tree oil to develop a biocompatible nanoformulation to deliver the herbal drug tanshinone-IIA (TSIIA), which is used for the treatment of acute lung injury (ALI), a severe condition often observed in patients with COVID-19. The nanoemulsion (NE) was optimised using ultrasound, and its efficacy was evaluated in an ALI model induced by lipopolysaccharides. Compared to free medication and blank-NE, TSIIA-NE demonstrated superior efficacy. Furthermore, the loading of TSIIA into the nanoemulsion formulation potentially amplified this effect, possibly due to the inherent pharmacological activities of tea-tree oil and rhamnolipids, as well as the improved in vivo performance of the nanoformulation [156].

In a study by Kubendiran et al. (2021), a natural biosurfactant extracted from *Lactobacillus casei* (MT012285) was used to develop a biosurfactant-based nano-topical ointment for wound treatment using a blend of *Tridax procumbens*-infused oil and gelatine-stabilised silver nanoparticles. The prepared ointment exhibited potent antimicrobial activity against clinical pathogens, including *S. aureus*, *E. coli*, *P. aeruginosa* and *K. pneumoniae,* with minimal haemolytic effects on red blood cells and low cytotoxicity on L929 fibroblastic cell lines. A wound scratch assay revealed a cell migration rate of 62% within 24 h, suggesting potential for topical wound treatment [157].

### 6.3. Liposomes

Biosurfactants have been increasingly utilised in the preparation of liposomes, which are lipid-based nanoparticles used for drug delivery and various biomedical applications, to replace PEG–lipids that may cause hypersensitivity reactions [158]. Glycolipids, such as rhamnolipids, in particular, can be employed to develop liposomes, as demonstrated with rhamnolipid-modified curcumin-loaded liposomes [159]. By incorporating biosurfactants into liposome formulations, several advantages can be achieved. Firstly, biosurfactants can enhance the stability and integrity of liposomes, optimise drug delivery efficiency, improve their shelf life and prevent aggregation [146,158,159]. Furthermore, biosurfactants can enhance the biocompatibility and biodegradability of liposomes, reducing potential toxic effects [159]. In a recently published study from 2022, researchers explored the use of sophorolipids for the modification of liposomes containing the nutraceutical drug pipeline. The study highlights the effectiveness of sophorolipids in enhancing the stability and dispersion of the liposomal formulation, while also improving the drug-loading efficiency [158].

## 7. Patents in the Biomedical and Pharmaceutical Fields Incorporating Biosurfactants

Patents related to the usage of biosurfactants in biomedical, pharmaceutical and related fields have been widely issued, indicating the growing interest and recognition of their therapeutic and functional properties. These legal protections not only provide recognition for the innovative efforts spent in developing biosurfactant-based solutions, but serve as a means for advancing research and development in the biomedical and pharmaceutical sectors, where the demand for novel, effective and safe therapies is paramount. Patents, therefore, provide a framework for incentivizing innovation, ensuring competition, and facilitating the translation of biosurfactant advancements from laboratory discoveries to impactful medical applications. These patents cover a range of topics, including biosurfactant formulation for purposes such as antimicrobial, antibiofilm, and coating agents, utilization in drug delivery systems, enhancements of pharmaceutical stability and bioavailability, as well as involvement in nanoparticle synthesis and modification for targeted therapies.

As confirmation of the interesting antiadhesive and antibiofilm activities of biosurfactants, several methodologies have been patented to covalently bind these molecules to the surfaces of various natures in order to prevent biofouling. The broad applicability and versatility of these natural surfactants extend their antiadhesive applications beyond the biomedical field, encompassing all sectors where microbial proliferation needs to be combated (ranging from agriculture and food industries to maritime transportation). In relation to applications employed to tackle microbial infection associated with the use of medical devices, methodologies for covalently linking biosurfactants to different materials have been described. A patent issued in 2022 describes protocols for coating polymeric and metallic materials with biosurfactants from *Pseudomonas aeruginosa*, *Bacillus amyloliquefaciens* and *Serratia marcescens* to inhibit biofilm formation by surface oxidation and the use of silane linkers [160]. In the same year, another patent was granted describing methods for coating medical devices with irregular and/or curved silicone surfaces via rhamnolipid grafting. This method involves different steps, such as another Brazilian patent describing a surfactin production process by isolates of *Bacillus subtilis* ATCC 19659 with anti-adhesive and antimicrobial properties against biofilm-forming pathogens on silicone-coated latex pieces [161].

Very recently, a patent concerning albofungin from *Streptomyces chrestomyceticus* BCC 24,770, issued as copolymer coating to inhibit fouling by marine microorganisms, demonstrated antibiofilm activity against ESKAPE bacteria [162]. In 2022, a patent was issued regarding medical devices, including voice prostheses, with irregular and/or curved silicone surfaces coated with rhamnolipids for the prevention or reduction of biofilm formation on the surface of the medical device. The method described involves the surface functionalization of silicone using the atmospheric or vacuum plasma discharge of argon with (3-Aminopropyl)triethoxysilane (APTES) or cyclopropylamine, followed by the covalent grafting of rhamnolipids using carbodiimide chemistry. The covalent grafting of rhamnolipids on APTES-modified silicone surfaces allowed for us to maintain the anti-biofilm activity up to 72 h with 80% reduction in *C. albicans* biofilm and more than 90% in *S. aureus* biofilm [163].

Another invention provides constituents and methods for treating, preventing, or disrupting biofilm formation on surfaces and in various bodily locations. The compositions utilize biological amphiphilic molecules produced by microorganisms (such as glycolipids, lipopeptides, flavolipids, phospholipids, fatty acid ether compounds, fatty acid ester compounds, and high-molecular-weight polymers), which can be combined with other biocidal substances such as antibiotics or essential oils to attack and weaken the bacterial biofilm matrix through enhancing the penetration of the biocidal substance. Such methods can be used to treat or prevent biofilm-related infections in various sites of the body and can also be applied to inert surfaces to inhibit the proliferation of biofilm-forming microbes [164]. Another Brazilian invention describes the production of surfactin by *Bacillus subtilis* isolates with anti-adhesive and antimicrobial properties against biofilm-forming isolates such as *S. aureus*, *S. epidermidis* and *E. coli* on stainless steel and titanium parts used in orthopaedic procedures [165]. A recent patent published in 2023 describes a method utilizing an extract containing biosurfactants derived from a bacterium, such as *Bacillus*, *Streptomyces*, *Mycobacterium*, *Micrococcus*, *Rhodococcus*, *Pseudomonas*, *Arthrobacter*, or *Staphylococcus*, as an effective antimicrobial agent against foodborne, plant bacterial or fungal pathogens [166].

In recent years, there has been growing interest in the use of probiotics that also possess the ability to produce biosurfactants. Patents have been filed to explore the potential of probiotics producing biosurfactants in improving health and wellbeing. By harnessing the synergistic effects of probiotics and biosurfactants, these patents aim to provide novel solutions for a wide range of health concerns, such as the enhancement of innate barrier functions in various areas of the body; the inhibition of pathogenic biofilm growth and adhesion; the promotion of commensal biofilm growth, treatment and prevention of fungal infections; the improvement in skin microbiome health; and the amelioration of skin barrier damage and inflammation, particularly in the context of inflammatory skin diseases like atopic dermatitis, psoriasis and acne [167,168,169,170,171].

As described in Section 5, biosurfactants have gained significant attention in the field of drug delivery, leading to their utilization in various patents. Overall, the incorporation of biosurfactants in patents for drug delivery holds great potential for improving therapeutic outcomes and expanding the possibilities of pharmaceutical formulations. A recently issued patent involves the development of nanostructured drug formulations that enhance both the effectiveness and bioavailability of the drugs, particularly in cancer treatment. These formulations utilize biosurfactants derived from microbial sources, replacing synthetic surfactants, to enhance the safety and efficacy of chemotherapy [172]. Another patent issued in 2023 provides a method for improving the penetration of a therapeutic agent into a microbial biofilm via the administration of a cavitation-enhancing agent, followed by exposure of the biofilm to at least one therapeutic agent. In this patent, it was demonstrated that rhamnolipids, in combination with antibiotics, have potent activity against persister cells and recalcitrant populations including anaerobically growing cells, non-respiring cells and small colony variants [173]. A patent issued in 2021 describes a combination treatment of rhamnolipids and niclosamide for non-alcoholic fatty liver disease; non-alcoholic steatohepatitis, including liver cirrhosis; and type 2 diabetes mellitus. By combining rhamnolipids with niclosamide, the water solubility of niclosamide is enhanced, leading to an increase in its oral bioavailability [174].

Biosurfactants have also emerged as valuable ingredients in the cosmetic field, offering unique properties and benefits for skincare, haircare and personal care products. Topical therapeutic compositions that utilize microbial biosurfactants have been patented for enhanced wound healing, scar reduction and an improvement in various skin conditions. These compositions and methods have been developed to reduce the healing time for skin wounds, diminish the appearance of scars, and provide benefits for conditions like acne, psoriasis and eczema [175]. In another patent, natural nano- or micro-emulsions containing rhamnolipids, mannosylerythritol lipids and the methods for producing them have been described for the treatment of the skin, hair follicles and related conditions as well as for the treatment of skin and hair pigmentation [176]. Cosmetic compositions and detergents containing biosurfactants have also been recently patented as environmentally friendly alternatives to cosmetic compositions based on non-renewable resources that may contain microplastic [177,178].

## 8. Conclusions and Future Perspectives

Interest and research continue to explore and optimize the utilization of biosurfactants for effective and eco-friendly pharmaceutical ingredients and biomedical components mainly due to their unique positive properties which meet the general environmental awareness, lifestyle and health aspirations and drive towards sustainability. Their wide-ranging potential applications vary from antimicrobial, antibiofilm, antiadhesive, antitumour, anti-inflammatory and antioxidant applications to possibilities of use in wound healing, immuno-modulation, probiotic and adjuvant capabilities, and nanoparticles and vaccine formulations for drug delivery. At present, biosurfactants remain to occupy a small market niche among surfactants due to their high production and purification cost; however, they are forecast to expand much further due to their increased functionality and synergistic interactions, which are helped by the commitment of some big industry players in the field. One of the main issues regarding the uptake of biosurfactants by the biomedical and pharmaceutical industries, in our opinion, is the purity of compounds generated, considering that most biosurfactants are typically produced as a combination of several congeners that can have different properties. The drive towards engineered strains for higher productivity or selectivity in product generation in addition to extensive toxicity testing and regulatory compliance are needed and will help future exploitation within the pharmaceutical, biomedical and health-related applications.

## Figures and Tables

**Figure 1 pharmaceutics-15-02156-f001:**
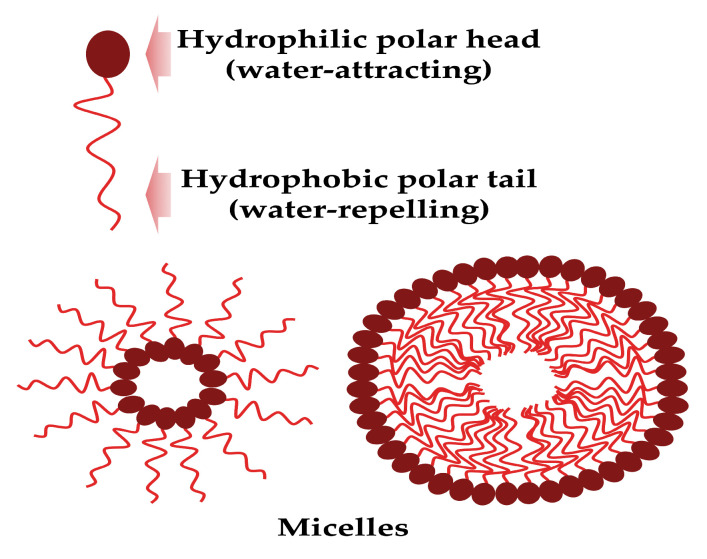
The biosurfactant general structure and micelle formation (created by PowerPoint © 2018 Microsoft).

**Figure 2 pharmaceutics-15-02156-f002:**
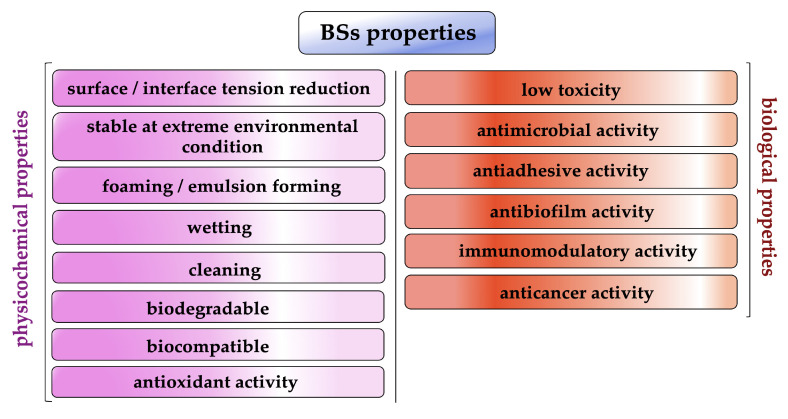
Biosurfactant properties (created by PowerPoint © 2018 Microsoft).

**Figure 3 pharmaceutics-15-02156-f003:**
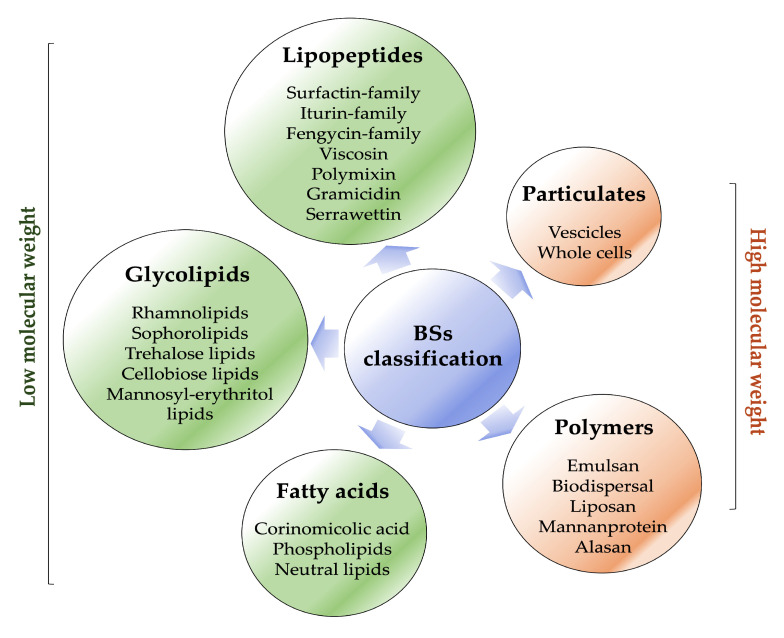
Biosurfactant classification (created by PowerPoint © 2018 Microsoft).

**Figure 4 pharmaceutics-15-02156-f004:**
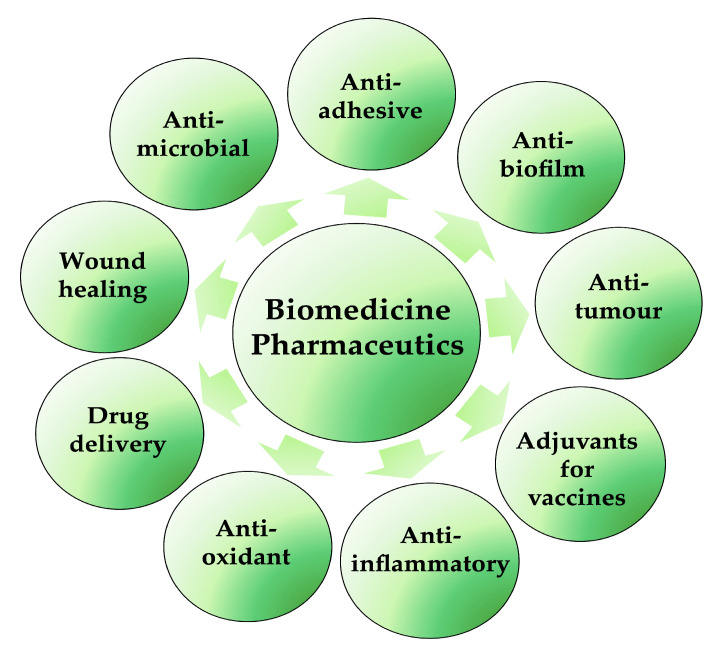
Biosurfactant applications in biomedical and pharmaceutical fields (created by PowerPoint © 2018 Microsoft).

**Figure 5 pharmaceutics-15-02156-f005:**
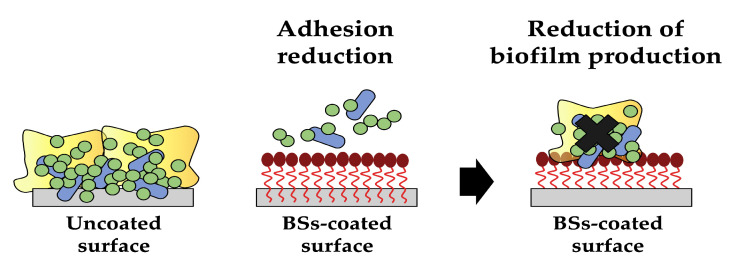
The antiadhesive/antibiofilm activity of biosurfactants (created by PowerPoint © 2018 Microsoft).

**Figure 6 pharmaceutics-15-02156-f006:**
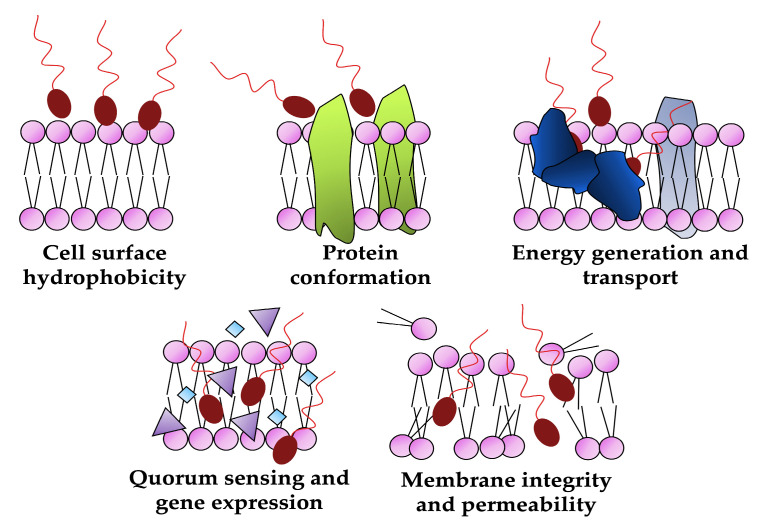
Biological targets of biosurfactants as antimicrobials (created by PowerPoint © 2018 Microsoft).

## Data Availability

The data presented in this study are available in this article.

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
