# Peer review of "Harnessing the Potential of Biosurfactants for Biomedical and Pharmaceutical Applications"

_pharmaceutics, 2023, doi:10.3390/pharmaceutics15082156_

Round 1

Reviewer 1 Report

Dear Authors,

The presented review manuscript entitled "Harnessing the Potential of Biosurfactants for Biomedical and Pharmaceutical Applications" presents very synthesized knowledge about the medical applications of biosurfactants. The subject of the manuscript is focused on the practical application of these compounds at the present moment as well as in the future perspective. The information is well presented and in my opinion could be very interesting for the potential reader. The bibliography list is sufficient and well organized. I have only one comment according to the antivirial properties of biosurfactants. These subject could be developped which is very important from the practical point of view.

Because of the above mentioned reasons in my opinion the manuscript could be accepted for the publication after the minor revision step.

Author Response

We thank the reviewer for their assessment of the paper and the favourable remarks. We agree on the significance of enhancing and expanding the section on the antiviral properties of biosurfactants. To address this, we have included some details about the effectiveness of sophorolipids against Herpes and HIV viruses. However, we had to limit the amount of material added taking into consideration concerns raised by the other reviewers regarding the overall length of the review.

Reviewer 2 Report

The present review focused on the potential application of biosurfactants from diverse microorganisms in facing medical issues, including infections/microbial biofilms, wound healing, cancer and immunological disorders. The incorporation of biosurfactant in delivery systems (such as nanoparticles, liposomes, emulsions) has also been described. The topic presented by the authors is certainly interesting and relevant. The review is well structed and written, and the available date in literature have been extensively described.

I have some minor comments:

-        Lines 86-87 “the use of non-harmful and eco-friendly products”. Actually there are some safety concerns about the use of biosurfactant from pathogenic/potentially pathogenic microorganisms. About the eco-friendly nature of biosurfactant (see also line 91) should be better explained, since often the protocol for biosurfactant isolation and purification require huge amount of water and/or organic solvent and can’t be considered eco-friendly.

-        Line 331: I think that the word “continue” can be removed.

-        Line 363: “Lactobacillus” in italics.

-        Figure 7: In my opinion can be omitted because is not so informative.

-        Lines 421-427: the sentence is very long and quite confusing to me; can be rephrased.

-        Line 427: “Bacillus subtilis” in italics.

-        Please insert the reference of program used for the creation of figures.

I would also suggest to insert one (or more) Table(s) to summarize the information reported in the text. For example a table reporting the source of biosurfactant (BS) (microorganism producer), the structure of BS, the concentration tested, the activity exerted (whether antbiofilms, antimicrobial, anticancer,…) and main finding, the experimental model. Another table summarizing the employment of BS in drug delivery system may improve the readability of manuscript as well.

The quality of English is good.

Author Response

We would like to thank the reviewer for their comments and careful revision of the manuscript. Below we respond to the various comments. Changes made in the revised manuscript are highlighted in yellow.

Q: Lines 86-87 “the use of non-harmful and eco-friendly products”. Actually there are some safety concerns about the use of biosurfactant from pathogenic/potentially pathogenic microorganisms. About the eco-friendly nature of biosurfactant (see also line 91) should be better explained, since often the protocol for biosurfactant isolation and purification require huge amount of water and/or organic solvent and can’t be considered eco-friendly.

A: Yes we acknowledged this point and integrated it as suggested.

Q: Line 331: I think that the word “continue” can be removed.

A: Yes acknowledged and removed.

Q: Line 363: “Lactobacillus” in italics.

A: Yes corrected.

Q: Figure 7: In my opinion can be omitted because is not so informative.

A: Yes this figure was removed.

Q: Lines 421-427: the sentence is very long and quite confusing to me; can be rephrased.

A: The sentence has been rephrased

Q: Line 427: “Bacillus subtilis” in italics.

A: Yes acknowledged and Changed.

Q: Please insert the reference of program used for the creation of figures.

A: Yes this was Inserted for all.

Q: I would also suggest to insert one (or more) Table(s) to summarize the information reported in the text. For example a table reporting the source of biosurfactant (BS) (microorganism producer), the structure of BS, the concentration tested, the activity exerted (whether antibiofilm, antimicrobial, anticancer,…) and main finding, the experimental model. Another table summarizing the employment of BS in drug delivery system may improve the readability of manuscript as well.

A: We align with the reviewer's observation that tables would effectively synthesize and elucidate the information provided in the text. While the idea is indeed noteworthy, given the extensive nature of the review, incorporating tables would expand the manuscript's length considerably.  Therefore, to address the reviewer's input, we have incorporated a referenced table within the text to highlight the main biosurfactant molecules and their corresponding microorganism producers.

Reviewer 3 Report

This review is a comprehensive report on the potential and applications of biosurfactant in medical field. It incorporated 175 references (among them at least 23 self-citations), the oldest one is from 2001. One of the major issue is, that almost the same authors published a similar structured review on the same filed in 2021 in the same journal. However, that report is neither cited, nor highlighted among the aims, why in such a short time two such similar reviews are written, are there any differences or the recent one is only the updated version etc.

Based on the keywords, there are not any differences. 

Since the paper is 27 page long, it is suggested to prepeare a Table-of-content like this:

1.     Introduction

2.     Biosurfactants: A real prospects for biomedical and pharmaceutical use?

3.     Biosurfactants for innovative coatings

4.     Biosurfactants as Biocontrol Agents

  4.1  Biosurfactants as antimicrobials

4.1.1. Marine microorganism

4.1.2. Probiotic Lactic Acid Bacteria

4.1.3. Other BSs producers

            4.2. Antiviral

4.2.1. SARS-CoV2 and biosurfactants

     5. Perspectives of biosurfactants for wound healing, anticancer and immune-modulatory applications

     5.1. Wound healing

     5.2. Anticancer agents

     5.3. Immuno-modulatory agents

6. Utilising biosurfactants as adjuvans in medicine: drug delivery systems

6.1. Nanoparticles

6.1.1. Nanoparticles with antibacterial activity

6.1.2. Nanoparticules for drug delivery

6.2. Microemulsions and nanoemulsions

6.3. Liposomes

7. Patents in the biomedical and pharmaceutical fields incorporating biosurfactants

8. Conclusions and future perspectives

Some remarks to the structure:  Despite it includes antiviral and antimicrobial effects, maybe would be beneficial to create antibacterial and antifungal paragraph over antiviral. 4.2.1. is a third level paragraph alon, there is no further (like 4.2.2. etc), so maybe it is not nessecary to make a sublevel for it.

The antibacterial part is a bit confusing, becasue Bacilli axample is under LAB (meanwhile generally Bacilli belongs to spore forming bacteria and LABs are not spore forming). However the cited reference deals with Lactobacilli produced biosurfactants and their medical aspects. 

Similar issue might be, that under 4.2.1. SARS-CoV2 subchapter a fungi produced protein is mentioned as an example (cyclosporin), but it is not a biosurfactant so far.

To summarize: despite this is a well established review its aims needs to be clarified regarding the previuos review; emphasis could be shifted to that fields which are new, thus overlapping can be decreased (for example almost the same Figures); its structure could be improved; It would be also beneficial to provide further clarification on why the emphasis is primarily on marine organisms and probiotic lactic acid bacteria as antimicrobial producers. Additionally, incorporating a few more examples of other biosurfactant antimicrobial producers would enhance the comprehensiveness and diversity of the presented research.   While bacteria, fungi and yeast were mentioned as biosurfactant producers, there were no specific examples provided for the pharmaceutical applications of biosurfactants derived from fungi and yeast.

Othr comments are in the attached version.

Author Response

We would like to thank the reviewer for their comments and careful revision of the manuscript. Below we respond to the various comments. Changes made in the revised manuscript are highlighted in yellow.

Reviewer comments

This review is a comprehensive report on the potential and applications of biosurfactant in medical field. It incorporated 175 references (among them at least 23 self-citations), the oldest one is from 2001.

Q: One of the major issues is, that almost the same authors published a similar structured review on the same filed in 2021 in the same journal. However, that report is neither cited, nor highlighted among the aims, why in such a short time two such similar reviews are written, are there any differences or the recent one is only the updated version etc.

Based on the keywords, there are not any differences.

A: We appreciate your concerns; we however, we would like to emphasize that our previous review had been referenced in the scope of the study: “The main objective of this review is to present a comprehensive update on the latest advancements, research findings, and insights into how biosurfactants can be utilized in the pharmaceutical and biomedical fields, building upon the previous literature output [12. Ceresa et al., 2021].” Nevertheless, we have made efforts to make the objectives of this new review more distinct and to highlight its differences from the previous one.

Q: Since the paper is 27 page long, it is suggested to prepare a Table-of-content.

A: Yes we acknowledged and integrated a table of content as suggested.

Q: Some remarks to the structure: Despite it includes antiviral and antimicrobial effects, maybe would be beneficial to create antibacterial and antifungal paragraph over antiviral. 4.2.1. is a third level paragraph alone, there is no further (like 4.2.2. etc), so maybe it is not necessary to make a sublevel for it.

A: Yes we acknowledged and Changed.

Q: The antibacterial part is a bit confusing, because Bacilli example is under LAB (meanwhile generally Bacilli belongs to spore forming bacteria and LABs are not spore forming). However the cited reference deals with Lactobacilli produced biosurfactants and their medical aspects.

A: The Bacillus example has been removed and a new appropriated example has been included.

Q: Similar issue might be, that under 4.2.1. SARS-CoV2 subchapter a fungi produced protein is mentioned as an example (cyclosporin), but it is not a biosurfactant so far.

A: Cyclosporin A has been removed and Sophorolipids included.

Q: To summarize: despite this is a well established review its aims needs to be clarified regarding the previuos review; emphasis could be shifted to that fields which are new, thus overlapping can be decreased (for example almost the same Figures); its structure could be improved; It would be also beneficial to provide further clarification on why the emphasis is primarily on marine organisms and probiotic lactic acid bacteria as antimicrobial producers. Additionally, incorporating a few more examples of other biosurfactant antimicrobial producers would enhance the comprehensiveness and diversity of the presented research. While bacteria, fungi and yeast were mentioned as biosurfactant producers, there were no specific examples provided for the pharmaceutical applications of biosurfactants derived from fungi and yeast.

A: Thank you for the feedback. Concerning the aim of the work, we have tried to elucidate the advancements and the new components incorporated compared to our previous review. We have also introduced some details on the significance of lactic acid bacteria (LAB) and marine microorganisms. Regarding the pharmaceutical applications of biosurfactants derived from fungi and yeast, we have integrated illustrative examples. We have also included the yeast Starmerella bombicola as a producer of sophorolipids, highlighting its multifaceted roles as antimicrobials, disruptors of biofilms, immunomodulators, and anti-inflammatory.

Q: Other comments are in the attached version.

We have modified text and figures according to the comments embedded in the text. Concerning Figure 4, we acknowledge the reviewer's concerns about its resemblance to the previous figure. This picture has been created to assist the reader in developing a conceptual map for the main subject of the review. Moreover, as noted by the reviewer, it introduces new insights in addition to those featured in the prior review.  The diagram highlights the key applications of biosurfactants in biomedical and pharmaceutical domains and we are of the opinion that all these applications should be incorporated, regardless of any prior presentations.

.